# Differences in levels of *E. coli* contamination of point of use drinking water in Bangladesh

**Md. Masud Hasan[1], Zahirul Hoque[2]\*, Enamul Kabir[3], Shahadut Hossain[4]**

**1** Research School of Social Sciences (RSSS) The Australian National University, Canberra, ACT, Australia, **2** United Arab Emirates University, Al Ain, United Arab Emirates, **3** University of Southern Queensland, Toowoomba, QLD, Australia, **4** Health Canada, Ottawa, Canada

\* Zahirul.Hoque@uaeu.ac.ae

**Data Availability Statement:** This study used data which is available on reasonable request from UNICEF MICS. All the personal identifiable information regarding the participants had been removed from the dataset. Informed consent was

## Abstract

This study aimed to quantify the inequalities and identify the associated factors of the UN sustainable development goal (SDG) targets in relation to safe drinking water. The concentration of the gut bacterium *Escherichia coli* in drinking water at the point of use (POU) and other information were extracted from the latest wave of the nationally representative Bangladesh Multiple Indicator Cluster Survey (MICS 2019). Bivariate and multivariable multinomial logistic regression models were used to identify potential predictors of contamination, whereas, classification trees were used to determine specific combinations of background characteristics with significantly higher rates of contamination. A higher risk of contamination from drinking water was observed for households categorized as middle or low wealth who collected water from sources with higher concentrations of *E. coli*. Treatment of drinking water significantly reduced the risk of higher levels of contamination, whereas owning a pet was significantly associated with recontamination. Regional differences in the concentrations of *E. coli* present in drinking water were also observed. Interventions in relation to water sources should emphasize reducing the level of *E. coli* contamination. Our results may help in developing effective policies for reducing diarrheal diseases by reducing water contamination risks.

## Introduction

Higher mortality rates from diarrheal diseases, predominantly in low- and middle-income countries, can be substantially reduced through interventions such as the provision of safe drinking water [1]. The availability of the latter for consumption is a human right [2], as reflected in the United Nations Sustainable Development Goals (SDGs) as safely managed targets [3–5] targets. Previous studies have shown that drinking water at the point of use (POU) is more likely to be contaminated than water collected from main sources, and, hence, the efforts required to achieve the SDG targets beyond the source infrastructure are more extensive and challenging [6–9]. In addition to the chemical and physical aspects, the quality of drinking water in relation to health issues may be assessed from the presence and concentration of microbial contamination [10, 11].

taken from participants before participating in the survey by the national statistical office, Bangladesh Bureau of Statistics and UNICEF. The data are available from the Bangladesh country survey conducted on 2019 through the survey page of the UNICEF through the link: https://eur03.safelinks. protection.outlook.com/?url=http%3A%2F% 2Fmics.unicef.org%2Fsurveys&data=04%7C01% 7Czahirul.hoque%40uaeu.ac.ae%7Cd3996d6279d 8412cb9bf08da1e211d9b%7C97a92b044c87434 19b08d8051ef8dce2%7C0%7C0%7C637855426 089569875%7CUnknown%7CTWFpbGZsb3d8ey JWIjoiMC4wLjAwMDAiLCJQIjoiV2luMzIiLCJBTiI 6Ik1haWwiLCJXVCI6Mn0%3D%7C2000&sdata= RBUI4yEBRcOpiomRIvzZ4xfHBTqgYw3dj1mgKBY mgaE%3D&reserved=0.

**Funding:** No funds, grants, or other supports were received.

**Competing interests:** The authors declare that they have no known competing financial interests or personal relationships which have, or could be perceived to have, influenced the work reported in this article.

The SDG target of "safely managed" water, in terms of being free of microbial contamination, can be assessed from the presence and concentration of the human gut bacterium *Escherichia coli* in drinking water. Based on the level of contamination, drinking water at source or at the POU is categorized with respect to potential health risks [12, 13]. To monitor the achievements of safely managed drinking water in relation to the SDG targets and develop related policy recommendations, recent literature was searched with the aim of determining the factors associated with *E. coli* concentration in drinking water. One of the major sources of contamination is the collection point where drinking water may have been exposed to microbes from the environment. Water collected from an "unimproved" source is more likely to be contaminated during extraction, which may become incorporated in POU drinking water [12, 14–17].

The type of main drinking water source and the level of contamination at the collection point have associations with the level of contamination at the POU. However, the association between improved water sources and the level of fecal contamination may not be decisive [18, 19]. Higher levels of fecal contamination in household water are associated with unimproved sanitation facilities [20]. If households use open defecation or unimproved facilities, the water source can be exposed to microbes from the excreta, and consequently, the quality of drinking water deteriorates. Recontamination of drinking water occurs as a result of poor management of household water resources [21, 22]. Increased risk of higher levels of contamination at the point of consumption is significantly associated with ownership of livestock [23].

A clear understanding of the health impacts of maintaining the quality of potable water can be achieved by passing on information to individuals through educational institutions and/or community initiatives. A positive association between hygienic water practice and the level of educational attainment has been reported in the recent literature [24]. For example, a study based on data from Ghana, Nepal and Bangladesh showed that ownership of any type of livestock was associated with an increased risk of fecal contamination of drinking water at the POU [25, 26]. Adequate water treatment methods at the POU can significantly reduce the presence and total counts of coliforms present in drinking water [27–29]. Socio-economic inequalities among households are usually reflected in access to quality livelihoods, especially in terms of accessing potable water. A higher asset index score measured in terms of household possessions was significantly associated with access to improved water sources and reduced *E. coli* contamination in the drinking water [30, 31]. In the literature, the general practice is to convert the concentration of *E. coli* in drinking water into categories of potential health risks. The standard procedure of measuring the association of categorical outcome variables with a set of covariates is to measure the adjusted odds ratio (AOR) in a multivariable analysis [12, 32–34].

There is a gap in the existing literature with regard to identifying the specific combinations of background characteristics associated with higher risks of *E. coli* contamination in POU drinking water in Bangladeshi households. To fill this gap, this study aimed at understanding the distribution of and differences among the level of *E. coli* contamination in different areas of Bangladesh. To accomplish the aim, unadjusted and adjusted associations between *E. coli* contamination and a set of covariates were estimated using bivariate and multivariable analyses. A machine-learning tool, the classification tree, was used to identify the distribution of *E. coli* concentration over interactions of the predictor variables. The analyses were conducted using the latest data of the nationally representative Multiple Indicator Cluster Survey (MICS) of Bangladesh conducted in 2019. Our results have implications for a better understanding of the SDG target 6.1 and for providing empirical evidence to support the development of feasible and effective plans to reach this target.

## Material and methods

### Data

This study was conducted using data from the latest wave of the Multiple Indicator Cluster Survey conducted in Bangladesh in 2019 (MICS 2019) [35]. The survey was designed to achieve reliable estimates at the national level different across urban-rural areas, administrative divisions and districts of Bangladesh. A two-stage, stratified cluster sampling technique was adopted for the purpose of survey implementation. The first stage sampling frame consisted of the primary sampling units (PSUs) obtained as the enumeration areas (EAs) based on the latest Bangladesh Population and Housing Census-2011. The main strata were defined as the urban and rural locations within each of the 64 districts. A probability proportional to size (PPS) sampling procedure was used to select the PSUs (3220) from each of the sampling strata. For each of the selected EAs, complete lists of households were prepared for the next stage of sampling. A systematic random sample of 20 households was drawn from each of the 3220 EAs selected in the first stage. From these selected households from each EA, four households were selected for assessing arsenic concentration in their drinking water. From the four selected households from each of the EAs, two households were randomly sampled for assessing *E. coli* concentration in household drinking water and at its "source". The sampling in these two stages was done using a systematic random sampling technique. Thus, the expected sample size of this study was 6440 households, which were selected for testing of *E. coli*. A total of 6069 (98.7%) of the selected households were successfully tested for both household and source water quality for *E. coli* concentration. A total of 17 cases, for which results were either lost or unreadable, were excluded from the study.

### Variables

**Outcome variable.** The dependent variable in this study was the quality of drinking water in terms of possible fecal contamination. The water samples were collected from each household by asking for "a glass of water that you would give a child to drink." The most recommended indicator for fecal contamination of drinking water is the number of bacteria counts of *E. coli* in a 100 ml sample of drinking water. For this purpose, 100 ml of sample water was filtered through a 0.45 μm filter (Millipore Microfil®, MilliporeSigma, Burlington, MA, USA) and placed onto a Compact Dry EC growth media plate (R-Biopharm AG, Darmstadt, Germany). After 24 hours of incubation at ambient temperature, the number of blue colonies was recorded. Household drinking water with less than one blue colony was termed "low risk", whereas those with 1–10 colonies were categorized as "medium risk", and samples with 11–100 colonies or more than 100 colonies were categorized as "high" or "very high" risk, respectively [36]. In this study, the last two categories were combined into a "high risk category".

**Predictor variables.** In our study, a set of predictor variables were used to test a possible association with the outcome variable. The choice of predictor variables was guided by the existing literature, knowledge of the researchers and the availability of information. *E. coli* contamination in the drinking water may have occurred at the point of water collection. The information about this variable was recorded and categorized in the same way as the outcome variable. The types of drinking water sources (categorized as "improved" and "unimproved") were considered as potential predictor variables. The locations of the drinking water sources may have been linked to water quality in two ways, as they were located in areas surrounded by cleaner environments, or they may have been contaminated through the process of carrying the drinking water back to the households. Based on the locations of the drinking water sources, households were categorized as those having the sources in dwelling/ premises or

elsewhere. The other predictor variables included in the analysis were: whether the water was treated or untreated, the type of toilet facility (improved and unimproved), place of residence and administrative division [37, 38]. Our study tested the hypothesis that the educational attainment of the head of the household influences the behavior of the household members when consuming safe drinking water. Based on their educational attainment, heads of households were categorized as either "no education" or "pre-primary", "primary" or "secondary level" of education or "higher level" of education. Several studies observed a positive association between ownership of livestock and contamination of water. According to the ownership status of any livestock, herds, other farm animals, or poultry, households were categorized as either "own" or "do not own" any of livestock. Our study also tested whether wealthy households, with better management, were able to keep contamination to a lower level. Based on whether households had greater ability to manage safe drinking water, the variable was categorized as either "a poor," "middle" or "rich" households.

## Statistical analysis

In determining the unadjusted association between *E. coli* contamination and selected background characteristics, a bivariate analysis was conducted. As the outcome variable and all covariates were categorical by nature, bivariate chi-square analyses were also carried out. In addition to the bivariate analysis, a multivariate analysis was applied in order to determine the adjusted association between the covariates and *E. coli* contamination in household drinking water. The outcome variable, *E. coli* contamination in household drinking water, had three levels, coded as 2 (for high concentration), 1 (for moderate concentration) and 0 (for low concentration). For a multivariable analysis with three levels of outcome variables, a multinomial logistic regression was employed and details of the model can be accessed in the existing literature [39–41]. This model has numerous applications in the domain of population health [42, 43].

In order to identify significant multifactor interactions of the covariates associated with the level of *E. coli* contamination, a classification tree was used. The methodology was guided by the conditional inference framework [44, 45]. Squared adjusted generalized variance inflation factor (GVIF) scores were used to quantify multicollinearity in the model [46]. Version 3.5.3 of the open-source software R [47] package and version 1.2–7 of the related partykit package [48] were utilized in order to analyze the data and to fit the models.

## Results

The results of the bivariate analysis showing the relationships between *E. coli* contamination in drinking water and the potential covariates are presented in Table 1. A higher level of contamination in the source of the drinking water resulted in a higher level of contamination at the POU, and this association was statistically significant ($p<0.001$). the type of the source of drinking water and the location of the facility were significantly associated with the level of *E. coli* contamination at the POU. *E. coli* contamination were significantly higher in water from unimproved sources than from improved sources. The contamination was significantly lower in water collected from sources located in households or premises than from those located outside. The results of the bivariate analysis also indicated that the proportion of households with *E. coli* contamination was lower for those using any water treatment methods. The percentages of households with higher levels of *E. coli* contamination were significantly lower in households with improved toilet facilities. Ownership of livestock significantly increased the likelihood of consuming drinking water with possible fecal contamination. Higher levels of education of the head of household resulted in lower levels of *E. coli* contamination in drinking

**Table 1. Percentage distribution of households with various level of *E. coli* contaminations in drinking water at the point of use (POU).**

| | *E. coli* in POU Drinking Water | | | Sample Size (N) |
|---|---|---|---|---|
| | Low | Moderate | High | |
| *E. coli in water source* (p < 0.001) | | | | |
| Low | 25.9 | 23.3 | 50.8 | 3741 |
| Moderate | 6.6 | 21.3 | 72.0 | 1326 |
| High | 4.4 | 6.8 | 88.8 | 985 |
| *Education of household head* (p < 0.001) | | | | |
| No or pre primary | 15.5 | 19.6 | 65.0 | 2142 |
| Primary | 17.4 | 19.4 | 63.1 | 1714 |
| Secondary or higher | 21.4 | 21.4 | 57.2 | 2196 |
| *Ownership of livestock* (p < 0.001) | | | | |
| No | 21.4 | 20.6 | 58.0 | 2354 |
| Yes | 16.1 | 19.9 | 64.0 | 3698 |
| *Household wealth status* (p < 0.001) | | | | |
| Poor | 15.0 | 19.7 | 65.3 | 2796 |
| Middle | 18.8 | 20.1 | 61.1 | 2350 |
| Rich | 26.3 | 21.9 | 51.9 | 906 |
| *Source of drinking water* (p < 0.001) | | | | |
| Improved | 18.6 | 20.3 | 61.1 | 5877 |
| Unimproved | 4.0 | 15.4 | 80.6 | 175 |
| *Location of drinking water source* (p < 0.001) | | | | |
| Dwelling/ Premises | 19.3 | 20.7 | 60.0 | 4373 |
| Elsewhere | 12.1 | 19.2 | 68.7 | 1260 |
| *Treatment of drinking water* (p = 0.002) | | | | |
| No | 17.6 | 20.2 | 62.1 | 5558 |
| Yes | 24.3 | 19.4 | 56.3 | 494 |
| *Type of toilet facility* (p < 0.001) | | | | |
| Improved | 19.0 | 20.5 | 60.5 | 5027 |
| Unimproved | 14.3 | 18.5 | 67.1 | 1025 |
| *Place of residence* (p < 0.001) | | | | |
| Rural | 17.1 | 20.1 | 62.8 | 4896 |
| Urban | 22.8 | 20.3 | 56.8 | 1156 |
| *Administrative division* (p < 0.001) | | | | |
| Barisal | 9.3 | 21.5 | 69.2 | 558 |
| Chittagong | 18.3 | 22.3 | 59.5 | 1046 |
| Dhaka | 16.9 | 16.6 | 66.6 | 1232 |
| Khulna | 15.3 | 19.9 | 64.8 | 947 |
| Mymenshing | 21.9 | 24.3 | 53.8 | 370 |
| Rajshahi | 21.9 | 15.4 | 62.7 | 764 |
| Rangpur | 24.5 | 24.8 | 50.7 | 747 |
| Sylhet | 18.8 | 21.4 | 59.8 | 388 |

water. Among the households categorized as poor, middle or rich, the proportion of sample households with low levels of *E. coli* contamination were 15.0%, 17.4% and 21.4% respectively. A higher percentage of households in rural locations (62.8%) consumed water with high *E. coli* concentrations than those in urban locations (56.8%). The percentage of households with high *E. coli* concentrations in drinking water was highest in Barisal Division, followed by Dhaka Division. The percentage was lowest in Rangpur Division, followed by Mymenshing Division.

The regional differences in the proportions of households with high levels of *E. coli* contamination were statistically significant.

A multivariable analysis using a multinomial logistic regression was employed to quantify the adjusted impacts of the covariates on *E. coli* contamination of drinking water with three levels. All of the significant variables in the bivariate analysis were included in the model. Because of possible multi-collinearity of the source and location of the drinking water, the source location of the latter was excluded from the model. The model outputs, along with the AOR and 95% confidence intervals (CI), are presented in Table 2.

**Table 2. Adjusted odds ratio (AOR), confidence intervals (CI) and *P*-values of moderate and high risk of *E. coli* contamination of drinking water at the point of use (POU) obtained from the multinomial logistic regression models.**

| | Moderate risk | | High risk | |
|---|---|---|---|---|
| | AOR (LCL–UCL) | P-Value | AOR (LCL–UCL) | P-Value |
| (Intercept) | 0.44 (0.29–0.67) | 0.000 | 0.37 (0.25–0.53) | 0.000 |
| *E. coli in water source* | | | | |
| Low | 1.00 | --- | 1.00 | --- |
| Moderate | 3.79 (2.92–4.92) | 0.000 | 6.32 (4.98–8.02) | 0.000 |
| High | 1.80 (1.19–2.71) | 0.005 | 12.92 (9.28–17.98) | 0.000 |
| *Ownership of livestock* | | | | |
| No | 1.00 | --- | 1.00 | --- |
| Yes | 1.17 (0.97–1.40) | 0.102 | 1.35 (1.15–1.58) | 0.000 |
| *Source of drinking water* | | | | |
| Improved | 1.00 | --- | 1.00 | --- |
| Unimproved | 2.51 (1.04–6.03) | 0.043 | 1.70 (0.75–3.85) | 0.206 |
| *Treatment of drinking water* | | | | |
| Yes | 1.00 | --- | 1.00 | --- |
| No | 1.44 (1.04–1.99) | 0.026 | 1.80 (1.35–2.40) | 0.000 |
| *Education of household head* | | | | |
| Secondary or above | | | | |
| No/Pre primary | 1.10 (0.89–1.37) | 0.370 | 1.18(0.98–1.42) | 0.083 |
| Primary | 0.98 (0.79–1.22) | 0.887 | 1.09(0.90–1.31) | 0.377 |
| *Type of toilet facility* | | | | |
| Improved | 1.00 | --- | 1.00 | --- |
| Unimproved | 1.01 (0.79–1.30) | 0.865 | 1.14 (0.92–1.42) | 0.214 |
| *Household wealth status* | | | | |
| Rich | 1.00 | --- | 1.00 | --- |
| Middle | 1.14 (0.87–1.49) | 0.335 | 1.43 (1.13–1.81) | 0.003 |
| Poor | 1.28 (0.95–1.73) | 0.106 | 1.74 (1.33–2.26) | 0.000 |
| *Place of residence* | | | | |
| Rural | 1.00 | --- | 1.00 | --- |
| Urban | 0.91 (0.72–1.15) | 0.429 | 0.85 (0.70–1.04) | 0.124 |
| *Administrative division* | | | | |
| Rangpur | 1.00 | --- | 1.00 | --- |
| Barisal | 2.55 (1.73–3.77) | 0.000 | 4.41 (3.10–6.26) | 0.000 |
| Chittagong | 1.16 (0.86–1.56) | 0.310 | 1.21 (0.93–1.59) | 0.154 |
| Dhaka | 1.03 (0.76–1.38) | 0.820 | 1.82 (1.41–2.40) | 0.000 |
| Khulna | 1.24 (0.91–1.69) | 0.164 | 2.06 (1.57–2.70) | 0.000 |
| Mymenshing | 0.97 (0.67–1.41) | 0.941 | 0.87 (0.62–1.22) | 0.418 |
| Rajshahi | 0.75 (0.54–1.03) | 0.073 | 1.54 (1.18–2.00) | 0.001 |
| Sylhet | 1.25 (0.84–1.84) | 0.255 | 1.59 (1.13–2.25) | 0.008 |

In the multinomial logistic regression model, a higher concentration of *E. coli* in the source of drinking water was significantly associated with a higher concentration *E. coli* in the POU drinking water. For example, households collecting water from a high-risk contaminated source were 11.92 (AOR: 12.92; 95% CI: 9.28–17.98) times more likely to have a high risk of *E. coli* contamination at the POU. On the other hand, for sources providing water with a moderate risk of contamination, the POU water was 5.32 (AOR: 6.32; 95% CI: 4.98–8.02) times more likely to have a high risk of *E. coli* contamination at the POU. Households collecting drinking water from unimproved sources were more likely to consume water with a higher level of *E. coli* contamination than those collecting from improved sources. The relationship was statistically significant for a moderate risk of contamination (AOR: 2.51, 95% CI: 1.04–6.03).

Households not treating their drinking water after collection were significantly more likely to consume water with a moderate ($p < 0.05$, AOR: 1.44; 95% CI: 1.04–1.99) or high risk ($p < 0.001$, AOR: 1.80; 95% CI: 1.35–2.40) of *E. coli* contamination. Ownership of a pet was significantly associated ($p<0.001$) with the consumption of water with a higher level of *E. coli* contamination. Compared with households not owning livestock, those owning livestock were 0.35 (AOR: 1.35; 95% CI: 1.15–1.58) times more likely to consume drinking water label as having a high risk of contamination. The other variable related to the household environment was the type of toilet facility available in the household. Not possessing a toilet facility or using an unimproved one was related to a higher risk of *E. coli* contamination of drinking water.

Wealth of household showed a consistent association with the level of *E. coli* contamination of potable water, though the relationship was more evident when considering the contamination in relation to higher risk. For example, with respect to rich households, middle and poor households were 0.43 (AOR: 1.43; 95% CI: 1.13–1.81) and 0.74 times (AOR: 1.74; 95% CI: 1.33–2.26) times more likely, respectively, to consume water with a high level of *E. coli* contamination. Urban households were less likely to consume *E. coli* contaminated drinking water at the POU, though the differences were not statistically significant. This study identified significant regional inequalities in *E. coli* contaminated water at the POU of household members. Households residing in Barisal and Khulna divisions were 3.41 (AOR: 4.41; 95% CI: 3.10–6.26) and 1.06 (AOR: 2.06; 95% CI: 1.57–2.70) times more likely, respectively, to use high *E. coli* contaminated drinking water compared to those residing in Rangpur Division.

The levels of contamination risks related to *E. coli* concentration in drinking water at the POU in Bangladesh for various combinations of the levels of background of households is presented in Fig 1. The root node of the classification tree is the level of *E. coli* contamination in the source of the water. This finding indicates that contamination of water at source was the major contributor to the risk of contamination in POU drinking water. Urban-rural residence did not appear to be a significant contributor to the outcome variable when the households collected drinking water with high level of *E. coli* contamination. The immediate left of the trunk is divided into branches based on the wealth of the households. For households consuming water from highly contaminated sources, the contamination level of the POU drinking water did not differ significantly between households with poor or middle incomes. For this group, the highest percentages of households (91.0%) had a high level of *E. coli* contamination in the POU drinking water (node 6). The third node (second node to the left trunk) indicates significant regional differences in the level of *E. coli* contamination (among the households categorized as rich and using drinking water from sources with high levels of *E. coli* contamination). The right trunk from the root node refers to households collecting drinking water with low or medium levels of *E. coli* contamination. All of the terminal nodes through this trunk (nodes 9, 11, 14, 15, 16, 18, 20 and 21) were related to relatively lower percentages of households with high level of *E. coli* contamination. From this group, a relatively higher proportion of households (67.0%) from Barishal Division had higher levels of *E. coli* contamination in

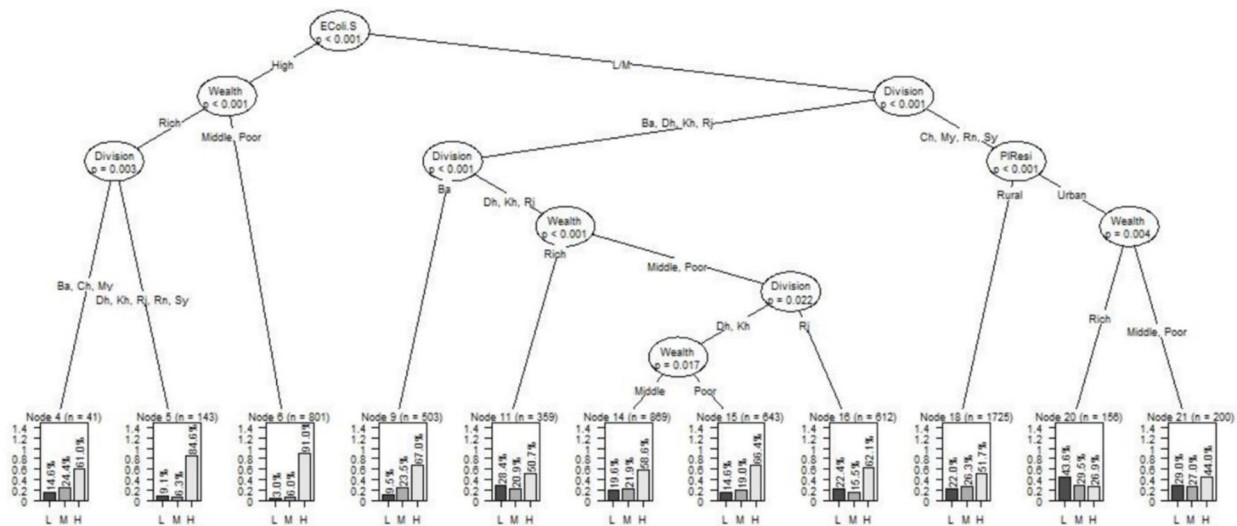

**Fig 1. Classification tree representing the distribution of levels of *E. coli* contamination of drinking water across different combinations of the levels of household characteristics.**

their drinking water (node 9). These proportion was followed (66.4%; node 15) by the poor households in Dhaka and Khulna Divisions and the middle/poor households from Rajshahi Division (62.1%; node 16).

## Discussion

This research quantifies the inequalities and identifies the associated factors of the SDG targets relating to the safe POU drinking water in Bangladesh based on microbiological quality. The results from of this study confirmed that the source of the drinking water did not have any significant association with the level of *E. coli* contamination in POU water and that using any water treatment method significantly reduced the concentration of *E. coli* concentration in POU drinking water.

This study also identified that a higher concentration of *E. coli* in the water source was significantly associated with the concentration of *E. coli* in the POU drinking water. This fact was supported by a number of studies conducted in Bangladesh and elsewhere [22, 49–52]. The positive association between the proportion of households with high concentrations of *E. coli* in POU water and also owning a pet might indicate the possibility of secondary contamination. Middle/poor households with high level of *E. coli* contamination at source had an approximately 90% risk of having a high level of contamination in their POU water.

The results of this study suggest that a higher proportion of *E. coli* contamination originated from the source of the drinking water. Emphasis should be put on improving the infrastructure of water sources in order to ensure that water is not being contaminated through the surrounding environment. Integrated plans need to be formulated by the central and local government and nongovernmental organizations (NGOs) in order to provide affordable and adequate water treatment facilities to the mass population. Significant regional differences in microbial contamination levels demands the adoption of alternative approaches at regional and local levels.

## Conclusions

Using nationally representative data and sophisticated statistical tools, this study identified associated factors of *E. coli* contamination in point of use (POU) drinking water in Bangladesh.

*E. coli* contamination of POU drinking water was significantly associated with the contamination of water at its source but not with the source type (improved or unimproved). Hence, the key factors of contamination at the source of collection should be identified and measures should be taken to avoid such contamination. Rural household should be educated regarding the possible recontamination of drinking water by livestock. Use of household water treatment facilities significantly reduces the *E. coli* contamination of drinking water, though the use of such facilities is limited. Integrated campaigns to promote the importance of treating water before drinking may raise the current rates of water treatment by households. Potential water treatment users should also be educated in how to use the methods more effectively in order to make water safe for drinking, and the materials for water treatment should be readily available. Significant regional differences in the levels of *E. coli* contamination in POU drinking water should be kept in mind in developing relevant policies. One of the limitations of our study was that the cross-sectional data were collected at a particular point in time, and so the seasonal effects on water quality were not assessed. Information regarding the distances of water sources from toilet facilities was also not available.

## Acknowledgments

UNICEF for Multiple Indicator Cluster Survey (www.childinfo.org) datasets, and the two anonymous referees for their valuable comments and suggestions.

## Author Contributions

**Conceptualization:** Md. Masud Hasan, Zahirul Hoque, Enamul Kabir, Shahadut Hossain.

**Data curation:** Md. Masud Hasan.

**Formal analysis:** Md. Masud Hasan.

**Writing – original draft:** Md. Masud Hasan, Zahirul Hoque, Enamul Kabir, Shahadut Hossain.

**Writing – review & editing:** Md. Masud Hasan, Zahirul Hoque, Enamul Kabir, Shahadut Hossain.

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
