## [Decision Letter · Decision Letter 0]

23 Dec 2021

PONE-D-21-23710Inequalities in E-coli contamination in the point of use drinking water in BangladeshPLOS ONE

Dear Dr. Hoque,

Thank you for submitting your manuscript to PLOS ONE. After careful consideration, we feel that it has merit but does not fully meet PLOS ONE’s publication criteria as it currently stands. Therefore, we invite you to submit a revised version of the manuscript that addresses the points raised during the review process.

We look forward to receiving your revised manuscript.

Kind regards,

Mentore Vaccari

Academic Editor

PLOS ONE

Journal Requirements:

Reviewers' comments:

Reviewer's Responses to Questions

**Comments to the Author**

1. Is the manuscript technically sound, and do the data support the conclusions?

Reviewer #1: Partly

Reviewer #2: Partly

2. Has the statistical analysis been performed appropriately and rigorously? 

Reviewer #1: Yes

Reviewer #2: Yes

3. Have the authors made all data underlying the findings in their manuscript fully available?

Reviewer #1: No

Reviewer #2: Yes

4. Is the manuscript presented in an intelligible fashion and written in standard English?

Reviewer #1: No

Reviewer #2: Yes

5. Review Comments to the Author

Reviewer #1: I think the topic is relevant and can broaden our knowledge on the situation related to SDG 6 in Asia. But more works to do by authors.

Major comments

- Please check the writing. The sentences are not well written as an academic journal. I strongly suggest authors to do a professional proofread before submit the revision.

- There are many repetition as well in the text.

- The story line of the paper is bad. Authors need to revise the whole draft.

- The way the authors write E. coli is not correct. Please revise the whole paper.

- There are many sentences that need references.

- After reading the paper, I guess the authors are not really familiar with the drinking water quality. E.g., writing E. coli is wrong, some terms are rarely used in the WASH sector (e.g., secondary contamination, source water). I suggest authors to read some papers related to WASH and water quality and use common terms in this field.

- Authors should add references from region outside Bangladesh, not only from Bangladesh. So the findings are relevant also to other countries/settings.

Please check specific comments in the pdf file. I put my comments there. Thank you and good luck for the authors!

Reviewer #2: General:

The manuscript is technically sound. Though it has used secondary data, the data was analyzed vigorously with an appropriate use of statistics. Data supports conclusions that have been drawn by the author. The study incorporated an appropriate number of samples, and the methodology used for statistical analysis is latest and appropriate for the study.

Major:

One of the aims of the paper is to monitor the empirical evidence to monitor the progress of the sustainable development targets in relation to the safe drinking water usage in Bangladesh. This aim was not achieved, or it is not clear how author intended to achieve this aim.

In introduction, it is not mentioned, why this study is new and different from the existing studies. Identifying the research gap is not clear. Suggest authors to enumerate more on literature to highlight the gaps in the current research studies and the novelty of this work.

In results (statistical analysis), Adjusted Odds Ratio (AOR) interpretation was not done correctly. It is not clear on what basis; the author subtracts 1 from AOR values. Authors need to use the AOR values or explain with clear base why the subtraction is needed.

In results, when discussing regional changes of E. Coli values, “Mymenshing” division values must be used as it is the only lower value than the “Rangpur” division value which is the default. Or explain that it has been disregarded as the values are not significant.

The conclusion is not addressed to the aims of the study fully.

Limitations of the study, recommendations to address for future works are not included which is also a major limitation of the paper.

Minor:

Abstract:

*2nd para, 1,2,3 lines. Using of “if” is not appropriately done which alter the idea of the sentence.

Introduction:

* 2nd para, 2nd line. "Or other" written in brackets, it's not clear which other microbes the author tries to specify. It is better to name a microbe or maybe a family of microbes.

* 2nd para, 12th line. Better to start "Type of ma...." as a new paragraph as it discusses a new point.

* 2nd para, 14th line. "improved sources may have faecal contamination above the WHO standard" here above, does not give the idea that improved sources have minimum vulnerability to contaminate.

* 2nd para, 20th line. Better to start "A clear understanding ..." as a new paragraph as it discusses a new point.

* 2nd para, 31st line. It is unclear who "respondents" refers to.

Variables: Predictor variables

* 1st para, 4th line. "may be carried through the source of water collection", should be corrected as "may be occurred at the source of water collection"

Statistical Analysis:

* 1st para, 14th line. 35,36 references should be superscript.

Results:

* 1st para, 2nd line. reference not available

6. PLOS authors have the option to publish the peer review history of their article (what does this mean?). If published, this will include your full peer review and any attached files.

Reviewer #1: No

Reviewer #2: No

---

## [Author Response · Author response to Decision Letter 0]

1 Mar 2022

Reviewers' comments to the author:

Reviewer #1: 

Overall comment: I think the topic is relevant and can broaden our knowledge on the situation related to SDG 6 in Asia. But more works to do by authors.

Response: The authors appreciate positive comments from the reviewer. The manuscript has been revised following the valuable comments and suggestions of the reviewer. 

Major comments

Comment: Please check the writing. The sentences are not well written as an academic journal. I strongly suggest authors to do a professional proofread before submit the revision.

Response: The manuscript has been proofread by a professional native English proofreader

Comment: There are many repetition as well in the text.

Response: the repetitions have been removed.

Comment: The story line of the paper is bad. Authors need to revise the whole draft.

Response: To improve the storyline, the whole manuscript has been revised carefully.

Comment: The way the authors write E. coli is not correct. Please revise the whole paper.

Response: The whole paper has been revised to replace E-Coli by E. coli.

Comment: There are many sentences that need references.

Response: References have been added as appropriate.

Comment: After reading the paper, I guess the authors are not really familiar with the drinking water quality. E.g., writing E. coli is wrong, some terms are rarely used in the WASH sector (e.g., secondary contamination, source water). I suggest authors to read some papers related to WASH and water quality and use common terms in this field.

Response: Revised.

Comment: Authors should add references from region outside Bangladesh, not only from Bangladesh. So the findings are relevant also to other countries/settings.

Response: Added

 

Please check specific comments in the pdf file. I put my comments there. Thank you and good luck for the authors!

Abstract: 

Revision: Just write this MICS 2019

Response: Revised.

Revision: I guess only the classification tree was used to find the combinations, not the regressions

Response: Revised.

Revision: What do you mean by secondary contamination?

Response: Revised.

Revision: Do you have a hypothesis that inequalities exist? I suggest to change word evident here.

Response: Revised.

Revision: water source not source water

Response: Revised.

Revision: The last sentence doesn't add something. it applies to all places in the world

Response: Revised.

Introduction:

Revision: in 2017, 1.6 million deaths globally including half a million children -> dont put it in the brackets, just write it in the text.

Response: Revised

Revision: just focus on SDGs and not MDGs

Response: Revised.

Revision: previous, not contemporary

Response: Revised.

Revision: the SDG targets beyond the infrastructure of source is more practical and challenging -> I dont understand why you write it more practical. 

Response: Revised.

Revision: health issue is not only about microbial, there are other chemical and physical aspects of water that also related to health

Response: Revised.

Revision: this is not the right way to write E. coli

Response: Revised.

Revision: unhygienic environment? this is the first time I heard this term. suggest to change this.

Response: Revised.

Revision: improved is only in MDGs. suggest dont use it here again.

https://washdata.org/monitoring/drinking-water

Response: Revised.

Revision: I never heard secondary contamination. suggest to change it to recontamination. 

Response: Revised.

Revision: suggest delete the words from "such as...". 

Response: Revised.

Revision: I am wondering why you have education/promotion here. this paragraph is too much information. the story line is also poor. you in the next sentence, your have livestock. why don't combine it with the sentences about secondary contamination above? 

I mean you can discuss first sources of contamination and then discuss WASH education/promotion/socio-economic determinants. now this paragraph looks messy.

Response: Revised according to the suggestion.

Revision: This paragraph should be in the methods, not introduction!

Response: Revised.

Revision: Do you use this bayesian? if not, why you mention it?

Response: Revised.

Revision: What is the knowlegde gaps?

Response: Revised.

Revision: repetition of the methods in the previous paragraph. 

Response: Revised.

Material and Methods 

Data

Revision: I guess all the procedure here is obtained from the MICS report? if yes, you must cite the source. Unless you did this study yourself.

Response: Cited

Variables

Outcome variable

Revision: This is not only in Bangladesh. WHO recommends in 100 ml sample in all over the world.

Response: Revised.

Revision: Again, I criticize a lot the storyline. why you have the sentence about the blue colonies here? it should appears after the sentence about incubation.

Response: Revised.

Revision: Please give references to this.

You can use this one: https://doi.org/10.3390/resources9120150

Response: Reference added

Predictor variables

Revision: Please put the references. maybe you can move the paragraph about factors associated with water quality in introduction to this section.

Response: Revised and reference added.

Revision: why you need to put administratif area as predictor? check this paper why they use region as predictor:

Understanding handpump sustainability: Determinants of rural water source functionality in the Greater Afram Plains region of Ghana

Response: Revised.

Revision: education has a significant relationship with water quality has been found in many studies. please find the citation.

Response: Revised.

Statistical Analysis 

Revision: bivariate effects? never heard this term

Response: Revised.

Revision: dont write it X2 (this is the test symbol), but chi-square (name of the test).

Response: Revised.

Revision: so what is the benefit of using bivariate tests?

Response: Revised.

Revision: multivariable approach? never heard this term. multivariate analysis

Response: Revised.

Revision: what variable? outcome?

Response: Revised.

Revision: three levels -> plural this sentence is not grammatically correct.

Response: Revised.

Revision: suggest to change trichotomous with multinominal logistic regression. Never heard trichotomous before. 

Response: Revised.

Revision: never heard most stable models -> is it stable?

Response: Revised.

Revision: so others methods are not meaningful? the word choice it not good.

Response: Revised.

Revision: suggest to remove this equation, unless you did economic study.

Response: Removed.

Results

Revision: what is this?

Response: Revised.

Revision: again story line!! put results about education together with other socio-economic variables (next paragraph). 

Response: Revised.

Revision: if not significant, then dont need to mention it

Response: Revised.

Revision: why you use Rangpur as the reference? please explain it in the methods

Response: Based on the prevalence.

Discussion

Revision: I think this study does not monitor the progress, because you use data which already presented in the MICS report.

Response: Revised.

Revision: I think you can remove this sentence. repetition of methods.

Response: Removed.

Revision: Please check my comments before. why then you use bivariate?

Response: Revised.

Revision: This first paragraph is repetition of introduction and methods! delete!

Response: Removed.

Revision: what do you mean by negative association? if they do HWT, the quality is poor? This is what I get from your sentence/writing. 

Response: Removed.

Revision: which findings? HWT and E. coli concentration? why you jump directly to diarrhea while you discuss HWT before?

Response: Revised.

Revision: not only in Bangladesh. in many studies as well: 

https://doi.org/10.2166/ws.2011.064

https://doi.org/10.1080/09603120410001725612

https://doi.org/10.3390/ijerph17072172

Response: References added.

Revision: we dont use improved water source in SDGs. so you cannot suggest this anymore. 

https://washdata.org/monitoring/drinking-water

Response: Revised.

Revision: how about the cost? Or my question is: is your recommendation possible, considering the cost?

Response: Revised.

Revision: you have too much recommendations! Please just give one (the most possible) recommendation, based on your findings. Not throw everything here. 

You also need to back up with references!

Response: Revised.

Revision: this last sentence is kind of empty. I mean of course you need to improve the quality to prevent diseases

Response: Revised.

Revision: Does you study has limitation or recommendation for future research?

Response: Revised.

Conclusion

Revision: major? It is a problem, but maybe not a major issue in of public health in Bangladesh. I mean maybe there are public health issues which are often discussed in the country rather than water quality. This kind of sentence needs a citation and should not put in the conclusion. 

Response: Revised.

Revision: you mean that your statistical analysis is able to find the causes? remember that regression is not able to find causality. 

check this in Google: "Eight myths about causality and structural equation models"

Response: Revised.

Revision: change to recontamination

Response: Revised.

Revision: water treatment facilities (centralised water treatment) or household water treatment? they are two different things. 

you write "significantly reduces" remember that you dont have any findings related to centralised water treatment (but you do recommend it in the discusion, which is fine). 

Response: Revised.

Revision: should also educate or should be educated?

Response: Revised.

Revision: this is sentence is not clear. what exactly should be keep in mind?

Response: Revised.

Reviewer #2: 

General:

The manuscript is technically sound. Though it has used secondary data, the data was analyzed vigorously with an appropriate use of statistics. Data supports conclusions that have been drawn by the author. The study incorporated an appropriate number of samples, and the methodology used for statistical analysis is latest and appropriate for the study.

Response: The authors thank the reviewer.

Major:

Comment: One of the aims of the paper is to monitor the empirical evidence to monitor the progress of the sustainable development targets in relation to the safe drinking water usage in Bangladesh. This aim was not achieved, or it is not clear how author intended to achieve this aim.

Response: The aim is revised. 

Comment: In introduction, it is not mentioned, why this study is new and different from the existing studies. Identifying the research gap is not clear. Suggest authors to enumerate more on literature to highlight the gaps in the current research studies and the novelty of this work.

Response: The research gap is mentioned in the revised manuscript. 

Comment: In results (statistical analysis), Adjusted Odds Ratio (AOR) interpretation was not done correctly. It is not clear on what basis; the author subtracts 1 from AOR values. Authors need to use the AOR values or explain with clear base why the subtraction is needed.

Response: The odds ratio for the reference category is 1.00. This one is subtracted from the other category to find the difference. 

Comment: In results, when discussing regional changes of E. Coli values, “Mymenshing” division values must be used as it is the only lower value than the “Rangpur” division value which is the default. Or explain that it has been disregarded as the values are not significant.

Response: The reference category is selected based on the output from the bivariate analysis.

Comment: The conclusion is not addressed to the aims of the study fully.

Response: Revised.

Comment: Limitations of the study, recommendations to address for future works are not included which is also a major limitation of the paper.

Response: Limitation is included in the revised manuscript.

Minor:

Abstract:

Comment: 2nd para, 1,2,3 lines. Using of “if” is not appropriately done which alter the idea of the sentence. 

Response: Revised

Introduction:

Comment: 2nd para, 2nd line. "Or other" written in brackets, it's not clear which other microbes the author tries to specify. It is better to name a microbe or maybe a family of microbes.

Response: Revised

Comment: 2nd para, 12th line. Better to start "Type of ma...." as a new paragraph as it discusses a new point.

Response: Revised

Comment: 2nd para, 14th line. "improved sources may have faecal contamination above the WHO standard" here above, does not give the idea that improved sources have minimum vulnerability to contaminate.

Response: Revised

Comment: 2nd para, 20th line. Better to start "A clear understanding ..." as a new paragraph as it discusses a new point.

Response: Revised

Comment: 2nd para, 31st line. It is unclear who "respondents" refers to.

Response: Revised

Variables: Predictor variables

Comment: 1st para, 4th line. "may be carried through the source of water collection", should be corrected as "may be occurred at the source of water collection"

Response: Revised

Statistical Analysis:

Comment: 1st para, 14th line. 35,36 references should be superscript.

Response: Revised

Results:

Comment: 1st para, 2nd line. reference not available

Response: Reference added.

---

## [Decision Letter · Decision Letter 1]

8 Apr 2022

Differences in levels of E. coli contamination of point of use drinking water in Bangladesh

PONE-D-21-23710R1

Dear Dr. Hoque,

We’re pleased to inform you that your manuscript has been judged scientifically suitable for publication and will be formally accepted for publication once it meets all outstanding technical requirements.

Kind regards,

Mentore Vaccari

Academic Editor

PLOS ONE

Reviewers' comments:

Reviewer's Responses to Questions

**Comments to the Author**

1. If the authors have adequately addressed your comments raised in a previous round of review and you feel that this manuscript is now acceptable for publication, you may indicate that here to bypass the “Comments to the Author” section, enter your conflict of interest statement in the “Confidential to Editor” section, and submit your "Accept" recommendation.

Reviewer #1: All comments have been addressed

Reviewer #2: All comments have been addressed

2. Is the manuscript technically sound, and do the data support the conclusions?

Reviewer #1: Yes

Reviewer #2: Yes

3. Has the statistical analysis been performed appropriately and rigorously? 

Reviewer #1: Yes

Reviewer #2: Yes

4. Have the authors made all data underlying the findings in their manuscript fully available?

Reviewer #1: No

Reviewer #2: Yes

5. Is the manuscript presented in an intelligible fashion and written in standard English?

Reviewer #1: Yes

Reviewer #2: Yes

6. Review Comments to the Author

Reviewer #1: Authors response well to my comments and have improved the draft. Thank you for all hard work. But since there are many new references in the draft, please check carefully all of them and make sure that the citation position/numbers arre correct.

Reviewer #2: It would be better if use percentages at least to interpret AOR values. Specially when you compares them to the standard value.

7. PLOS authors have the option to publish the peer review history of their article (what does this mean?). If published, this will include your full peer review and any attached files.

Reviewer #1: No

Reviewer #2: No

---

## [Editor Report · Acceptance letter]

20 Apr 2022

PONE-D-21-23710R1 

Differences in levels of *E. coli* contamination of point of use drinking water in Bangladesh

Dear Dr. Hoque:

I'm pleased to inform you that your manuscript has been deemed suitable for publication in PLOS ONE. Congratulations! Your manuscript is now with our production department. 

Kind regards, 

on behalf of

Professor Mentore Vaccari 

Academic Editor

PLOS ONE